

# The EarthCARE Mission: Science Data Processing Chain Overview

Michael Eisinger[1], Fabien Marnas[2], Kotska Wallace[2], Takuji Kubota[3], Nobuhiro Tomiyama[3],
Yuichi Ohno[4], Toshiyuki Tanaka[3], Eichi Tomita[3], Tobias Wehr[2,†], and Dirk Bernaerts[2]

[1]European Space Agency, ESA-ECSAT, Fermi Avenue, Didcot OX11 0FD, United Kingdom
[2]European Space Agency, ESA-ESTEC, Keplerlaan 1, 2201 AZ Noordwijk, The Netherlands
[3]Japan Aerospace Exploration Agency (JAXA), 305-8505 2 Chome-1-1, Sengen, Tsukuba, Ibaraki, Japan
[4]National Institute of Information and Communications Technology (NICT), 184-0015 4 Chome-2-1, Nukui Kitamachi,
Koganei, Tokyo, Japan
[†]deceased, 1 February 2023

**Correspondence:** Michael Eisinger (Michael.Eisinger@esa.int)

**Abstract.** The Earth Cloud Aerosol and Radiation Explorer (EarthCARE) is a satellite mission implemented by the European Space Agency (ESA) in cooperation with the Japan Aerospace Exploration Agency (JAXA) to measure global profiles of aerosols, clouds and precipitation properties together with radiative fluxes and derived heating rates. The data will be used in particular to evaluate the representation of clouds, aerosols, precipitation and associated radiative fluxes in weather forecasting
and climate models.

The science data acquired by the four satellite instruments are processed on ground. Calibrated instrument data – level 1 data products – and retrieved geophysical data products – level 2 data products – are produced in the ESA and JAXA ground segments. This paper provides an overview of the data processing chains of ESA and JAXA and explains the instrument level 1 data products and main aspects of the calibration algorithms. Furthermore, an overview of the level 2 data products, with
references to the respective dedicated papers, is provided.

## 1   Introduction

The Intergovernmental Panel on Climate Change (IPCC, 2023) has recognised that:

> While major advances in the understanding of cloud processes have increased the level of confidence and de-
> creased the uncertainty range for the cloud feedback by about 50 % compared to AR5, clouds remain the largest
> contribution to overall uncertainty in climate feedbacks (high confidence).

The Earth Cloud, Aerosol and Radiation Explorer (EarthCARE) mission will allow scientists to address this uncertainty, enabling a direct verification of the impact of clouds and aerosols on atmospheric heating rates and radiative fluxes. The collection of global cloud, aerosol and precipitation profiles, along with co-located radiative flux measurements, will be used to evaluate their representation in weather forecast and climate models, with the objective to improve parameterisation schemes. Earth-
CARE is an ESA Explorer mission that is being carried out in a collaboration with JAXA, which is providing one of the instruments. A mission overview by Wehr et al. (2023) describes the mission, its science objectives, observational require-



ments, ground and space segments. EarthCARE will fly in a sun-synchronous, low-Earth orbit, with a Mean Local Solar Time of 14:00 ±5 minutes and a 25-day repeat cycle. It will fly at a rather low altitude in order to maximise performance of the active instruments with respect their power needs. The satellite embarks four instruments: the ATmospheric LIDar (ATLID), the Cloud Profiling Radar (CPR), the Multi-Spectral Imager (MSI) and the Broad-Band Radiometer (BBR). The UV lidar, equipped with a high-spectral-resolution receiver and depolarisation measurement channel, will measure vertical profiles of aerosols and thin clouds and allow for classification of five aerosol types. The highly sensitive 94 GHz (W-band) cloud radar will provide measurements of clouds and precipitation. Its sensitivity partly overlaps with the lidar but its signal can penetrate through, or deep into, the cloud, beyond where the lidar signal attenuates. Furthermore, it has Doppler capability, allowing measurement of the vertical velocity of cloud particles. The imager has four solar and three thermal channels that will provide across-track swath information on clouds and aerosols, and facilitate construction of 3D cloud-aerosol-precipitation scenes for radiative transfer calculations. The Broad-Band Radiometer measures solar and thermal radiances in three fixed viewing directions, along the flight track, which will be used to derive the top-of-atmosphere flux estimates. ATLID and CPR data will be used to calculate atmospheric heating rates and radiative fluxes, using 1D and 3D radiative transfer models, which results can be compared against the flux estimated from EarthCARE Broad-Band Radiometer measurements.

Science data acquired by the instruments will be processed in ground segments at both Europe and Japan. As well as the four level 1 instrument data streams, a particular aspect of the EarthCARE mission is the large number of level 2 data products that will be generated in order to retrieve a variety of geophysical data products (see section 8.2 in Wehr et al. (2023) for a description of product levels). These will take advantage of the synchronous data from multiple instrument sources, as well as making use of a dedicated, auxiliary stream of meteorological data that will be provided from the European Centre for Medium Range Weather Forecasting (ECMWF). The purposes of the data products cover, for example, target classification, vertical profiles of microphysical properties of ice, mixed and liquid clouds, particle fall speed, precipitation parameters, and aerosol type. This paper gives an overview of the processing chain development, the scientific data products and their retrieval algorithms, simulation and testing. Section 2 provides an overview of the processing chains at ESA and JAXA, illustrating the production model. Section 3 is dedicated to the level 1 data processors and products for the four EarthCARE instruments. Section 4 describes the auxiliary data processors and products, which make use of meteorological data from ECMWF and provide a common spatial grid for synergistic processors. Section 5 describes the level 2 processors and products, developed in Europe, Japan and Canada. Section 6 gives an overview of the development that has been undertaken for the processing chain, end-to-end simulator (E3SIM), generation and use of test data sets, and some of the lessons that have been learnt. Section 7 provides useful information concerning product format and conventions.

## 2   Processing chains

Instrument data is transmitted in Instrument Source Packets (ISP), from the satellite to ground, via the satellite's X-band antenna and enters the Payload Data Ground Segment (PDGS). As one of its responsibilities the PDGS processes the ISP data to release a range of data products. It also monitors the data quality and performs the mission planning, including for





instrument calibration activities. PDGS initial processing takes ISPs and marshals them into a consecutive time sequence in an individual level 0 product for each instrument, annotated with ancillary data that is also contained in the ISPs. The instrument level 1 processors take as their input these level 0 products and create level 1b products that are fully calibrated and geolocated instrument science measurements on the native instrument grid for ATLID, MSI and the BBR single pixel product. The nominal BBR product is integrated 10 km along track. Additionally an MSI level 1c product is produced, with interpolation to a spatial grid common to all MSI bands. CPR level 1 data is produced by JAXA and made available at the PDGS for use in the level 2 processors.

Additionally there are two auxiliary products that are introduced into the processing chain. These are ECMWF meteorological fields limited to EarthCARE swath and a spatial grid shared by all instruments ("joint standard grid").

The PDGS performs an instrument data calibration and monitoring function at the Instrument Calibration and Monitoring Facility (ICMF). Data, generally at L1, from routine on-orbit calibrations of the three European instruments is plotted and reviewed at pre-specified intervals, in order to assess whether it is necessary to, for instance, modify a parameter (set) in the Calibration and Characterisation Database (CCDB) or perform an additional calibration. During the mission's operating phase, data quality will also be monitored by a team, with members from ESA and the science user community, that assesses the geophysical data products.

## 2.1    ESA processing chain

The complete set of ESA data products, plus JAXA's CPR level 1b product, is generated according to the ESA EarthCARE Production Model, depicted in Fig. 1. It includes level 0, level 1 and level 2 data products, as well as auxiliary and supporting data products. The processing chain is run at the PDGS, with two exceptions: the CPR level 1b product (section 3.2) is generated in the JAXA ground segment using a data processor developed by JAXA, and the X-MET data product (section 4.1) is produced at ECMWF using a data processor developed by ESA.

Level 2 products are distinguished between level 2a (L2a) and level 2b (L2b). L2a refers to a data product derived from only one EarthCARE instrument, for example, ATLID. L2b refers to a data product derived from two or more instruments, for example, AC-TC is the target classification (TC) synergistically derived from ATLID (A) and CPR (C).

Every data product is produced by a processor. A processor either produces one data product or more than one data product. If a processor produces only one data product, the name of the processor and the name of the data product is the same. For example, the C-CLD processor produces only the C-CLD data product and the A-FM processor produces only the A-FM data product. If a processor produces more than one data product then it has a name different from the products. For example, the A-PRO processor produces the data products A-AER, A-ICE, A-TC and A-EBD. Fig. 1 gives an overview of all ESA processors and data products, including their inter-dependencies. If several data products are produced by one processor, this is indicated by a solid line around the respective data products, with the name of the processor given, for example, C-PRO and A-PRO. If a processor produces only one data product, the processor name is not explicitly indicated, as it is the same as the data product's name. For the data product user, the name of the processor is, however, not relevant. All data products can be fully identified by their data product name only.







**Figure 1.** The ESA "EarthCARE Production Model" shows all ESA data products and the CPR level 1b product (C-NOM), which is produced by JAXA. Level 1 products (L1b, L1c/d) and auxiliary data (Aux) products are described in sections 3 and 4 respectively. Level 2 products and their retrieval algorithms (L2a, L2b) are described in this AMT Special Issue according to Table 1 (L2a) and Table 3 (L2b).

## 2.2 JAXA processing chain

The Production Model for JAXA products is illustrated in Fig. 2, showing the flow of the processors/algorithms from L1 to L2 products. Level 2 (L2) is categorized into L2a and L2b with the same definition as ESA products; L2a products are those retrieved from a single instrument, while L2b products are those derived from multiple instruments.

JAXA data products are categorized into two groups: Standard Products and Research Products. All products fall into one of the two categories, depending on the maturity of their algorithm development. Standard products (Research products) are

indicated in solid-line (dashed-line) boxes in Fig. 2, respectively. The differences in the Standard Product and the Research Products are;



– Standard Product

    1. The algorithms in the Standard Products are relatively mature and have heritage from past studies.

    2. Standard products are strongly promoted to be developed and released.

3. Products are processed in the JAXA Mission Operation System and released from JAXA G-Portal website (https://www.gportal.jaxa.jp), together with the standard products of other JAXA Earth observation satellite missions.

– Research Product

    1. The algorithms in the Research Products consist of new research developments that are challenging, yet scientifically valuable.

2. Research products are promoted to be developed and released.

    3. Products will be processed and released from JAXA Earth Observation Research Center (https://www.eorc.jaxa.jp/EARTHCARE/index.html) and/or Japanese institute/universities.

## 3   Level 1 data processors and products

### 3.1   ATLID

The High Spectral Resolution Lidar, ATLID, will measure vertically resolved profiles (pure atmospheric and ground-backscattered optical power) over 3 optical paths. In order to increase the signal to noise ratio, it is possible to integrate two consecutive lidar shots.

    1. Mie co-polar channel: which is the backscattered signal, co-polarised with respect to laser transmission, received within the transmission bandwidth of the High Spectral Resolution Etalon (HSRE). It represents mostly the particles signature, with narrow spectral broadening.

    2. Rayleigh (co-polar) channel: which is the backscattered signal, co-polarised with respect to laser transmission, reflected by the HSRE. It represents mostly the molecules signature, with Rayleigh broadening.

    3. Cross-polar channel: which is the total backscattered signal (Mie + Rayleigh), cross-polarised with respect to laser transmission.

The ISP (Instrument Source Packets) generated by ATLID contain the measurement data as well as ancillary data such as number of accumulated shots, instrument mode, time information, laser frequency, laser energy, detector saturation, efficiency of the coalignment loop etc. ISPs are transmitted from the satellite to the PDGS, where the L0 data product is generated.

The ATLID ECGP (EarthCARE Ground Processor) is the L1 processor which ingests and processes the L0 files generated from the ISPs. At level 1, the main ATLID product is range-corrected attenuated backscatter signal that is the product between





# EarthCARE JAXA L2 Production Model



**Figure 2.** The JAXA "EarthCARE Production Model" shows all JAXA data products and ESA's level 1 and Aux products. Level 2 products and their retrieval algorithms (L2a, L2b) are described in this AMT Special Edition according to Table 2 (L2a) and Table 4 (L2b).

the backscatter coefficient and the round-trip atmospheric transmission versus altitude. Indeed, the signals at instrument output (i.e. at entrance of CCD) are a combination of the backscatter signals at instrument input (i.e the pure contribution from the atmospheric Mie co-polar, Rayleigh co-polar, and cross-polar optical power) after atmospheric effects. More precisely, the instrument output signals are linked to the input signal (after subtraction of the background) via a linear combination (which can be expressed in a matrix form) that describes the optical chain (transmissions of the different optical components,

including cross-talks) and the detection chain. The major purpose of the L1 processor is to invert this matrix in order to retrieve the range-corrected attenuated backscatter signal. These variables will be contained in the nominal (L1B-NOM) products from EarthCARE. In addition, a number of L1 calibration products will be generated.





The L1 processor parameterisation relies on several CCDB (Characterization and Calibration Database) files, containing the processing parameters required by the processor and describing either the payload behaviour or the algorithm settings. The CCDB is initially populated with data from ATLID on-ground characterisation and calibration. During flight it will be modified when necessary, updating parameters on the basis of on-orbit calibrations. The frequency of a parameter update extends from almost-fixed (for a number of on-ground characterised parameters) to regularly updated (for the outputs of the in-flight calibrations).

The ATLID modes of operation supported by the ECGP consist of the nominal lidar measurements (or NOM) interspersed with different calibration operations:

1. **FSC: Fine Spectral Calibration**: consists of a fine frequency tuning of the laser around the current operational laser frequency, in order to minimize the spectral cross-talk on the Rayleigh channel. The optimal frequency is then selected for nominal measurements - performed once a week.

2. **DCC : Dark Current Calibration**: consists of measuring the frequency-resolved dark current on each channel, by measuring the detectors signals while shutting-off the laser emitter - performed once a month.

3. **CSC : Coarse Spectral Calibration**: consists of tuning the frequency of the laser emitter (by relatively large frequency increments), in order to minimize the spectral cross-talk on the Rayleigh channel. A set of four to five optimal frequencies are chosen (the choice of the frequency is then done off-line with respect to other instrumental criteria) - performed at the beginning of life and when needed.

Therefore, the complete list of ATLID level 1b products is the following:

1. **A-NOM**: the nominal L1B-data : geolocated, calibrated range-corrected attenuated backscatter signal profiles, along with segregated-type associated errors and with intermediate products from the processing (raw signals, offset-removed signals, background-removed signals, energy-normalised signals and relative signals.)

2. **A-FSC**: the Fine Spectral Calibration products : optimal frequency and associated cross-talk, plus quality criteria (and intermediate products : geolocation of the samples, raw signals and retrieved spectral cross-talks for each point.)

3. **A-DCC**: the Dark Current Calibration products : DSNU (Dark Signal Non Uniformity) maps for the three channels and associated errors and quality criteria (and intermediate calculations : geolocation of the samples, raw signals, offset as a function of vertical sample for each channel.)

4. **A-CSC**: the Coarse Spectral Calibration products : frequency of each (4 or 5) optimum point, the associated cross talks (and intermediate products : geolocation of the samples, raw signals and retrieved spectral cross-talks for each point.)

The following sections describe the processing sequence.



### 3.1.1 Identification and counting of measurement data packets

Using ancillary data, the L1 processor identifies the packets that contain invalid raw data. The validity of these data has been assessed on-board during in-flight operations. The packets are accepted with respect to several criteria: supported instrument mode (measurement validity bit), emitted beam quality (spectral emission quality), control loop status (co-alignment emitter/receiver) detection health (background and signal saturation).

### 3.1.2 Geolocation

1. Range Evaluation Location: range information give the distance from the receiver for each echo sample. Range computations are independent of instrument mode and do not use measurement data stream, they rely only on the integration time and accumulation factor for the high (0 to 20km) and low (20 to 40km) resolution part of the echo signal and the computer clock.

2. Samples geolocation: satellite position, altitude and pointing data allow to perform geolocation of atmospheric echoes. The temporal attitude and orbital information is provided as auxiliary input files (line of sight is provided in a CCDB file) to the software that will compute the following steps for each profile time stamp :

   – Determination of position and attitude from time stamp;

   – Computation of line-of-sight targets;

   – Computation of geodetic coordinates of samples;

   – ECEF (Earth Centered Earth Fixed) frame to geodetic coordinates transformation.

3. Calculation of atmospheric parameters associated to the samples: using the geolocation information, atmospheric temperature and pressure are attributed to each sample by interpolation (spatial and temporal) of the ECMWF forecast data contained in the X-MET product. Besides delivering this information in the output product, these variables are used to calculate the effective Mie spectral cross-talk (contamination from the Rayleigh channel into the Mie channel) as the broadening of the Rayleigh backscatter depends on temperature and pressure profiles.

### 3.1.3 Radiometric pre-processing

1. Detector voltage offset correction: The signal offsets are continuously assessed and updated from detection raw data. These offsets come from the last analogue stage in the ATLID detection chain and are very slow varying perturbations of the detection signal, linked to laser transmitter thermal drifts. These offsets are evaluated in the first samples, before echo acquisition, and subtracted from the measurement.

2. Correction of dark signal non-uniformity: In this step, the dark signal is subtracted from the science data. The sample-resolved dark signal maps (Dark Signal Noise Uniformity (DSNU) maps) per channel are extracted from the last maps contained in the CCDB (i.e those produced during the last Dark Current Calibration (DCC) sequence).





3. Background subtraction: The background radiation needs to be subtracted from the atmospheric profiles. To do so, the background is measured just before (first sample) and after (last sample) echo acquisitions at each shot. A linear interpolation of both measurements is performed to retrieve background level in each vertical echo sample during acquisition. The background is then subtracted for each sample.

4. Laser energy normalisation: In order to mitigate the impact of laser energy fluctuation, a normalisation is performed. For this purpose, two reference energy levels are reference in the CCDB: a threshold energy as well as a reference energy. This normalisation is performed in two steps, (a) the mean energy over the accumulated shots is compared to the threshold energy (and the profile is discarded if lower than the threshold), (b) for the processed data, each profile is normalized by the reference energy.

### 3.1.4  Cross-Talk management

1. Cross-Talk analysis: There are a number of cross-talk parameters that should be considered: spectral cross-talk that is contamination of Mie channels into the Rayleigh, and vice-versa, plus polarisation cross-talk that is contamination between the co- and cross-polar channels. The objective of spectral cross-talk calibration is to compute cross-talk parameters that enable to retrieve, up to an absolute constant factor, Mie and Rayleigh scattering profiles from corrected raw signals delivered by the instrument. Absolute lidar constants, computed from on-ground absolute calibration process, must then be used to retrieve the input signal. Relevant information is contained in the CCDB. The initial cross-talk parameters are also computed on-ground during a specific calibration stage. In-flight calibration is carried out constantly during the measurements mode. This is performed in two successive steps:

   (a) **Rayleigh spectral cross-talk** (contamination from the Mie into the Rayleigh channel). The baseline method for inferring the Rayleigh cross-talk value is to use ground echoes that can be assimilated as pure Mie signals (such as deserts or ice covers). Over these identified areas, the signals of the Rayleigh and Mie channels are averaged to reach a sufficient signal-to-noise ratio. The cross-talk value is then inferred by computing the ratio of Rayleigh and Mie channels. In order to increase the number of values, dense clouds are also used as determination targets. The method is based on iteratively determining the cross-talk value leading to the smoothest path for the Rayleigh signal within the cloud. The cross-talk values are then interpolated along-track from these two series of anchor points.

   (b) **Mie spectral cross-talk** (contamination from Rayleigh into the Mie channels). The method to infer the Mie cross-talk value relies on computing the ratio between signals (after radiometric pre-processing) on Mie channel and Rayleigh channel for high atmospheric layers (samples higher than 30 km of altitude). We suppose that at these high-altitudes, only Rayleigh scattering occurs (i.e., we have a pure Rayleigh spectrum). These signals are averaged along a determined number of samples along-track (estimation window) to reach a sufficient signal to noise ratio. The Mie cross-talk is a quantity which varies with altitude since molecular broadening is temperature and pressure dependent: the width of the Rayleigh signal varying with the atmospheric conditions, the proportion of Rayleigh





signal contaminating the Mie channel will then vary accordingly. To take that effect into account, the cross-talk
       is first estimated at a reference altitude (at which a reference temperature is estimated) and the variation of the
       cross-talk along the vertical axis is then computed from a 2-dimensional (T, p) look-up table as a function of
       the temperature and pressure profiles interpolated from the X-MET product for each shot. The cross-talk values
       are finally interpolated from the values determined in each estimation window. Following the same approach, the
Rayleigh and Mie channel lidar constants are determined in each window.

2. Channel demultiplexing. This part of the processing consists of three successive steps:

   (a) Construction of the vertical profile of Mie cross-talk value: based on the Mie cross-talk value inferred at the pre-
       ceding step, and its associated temperature, as well as the temperature-dependent variation law and the interpolated
       temperatures of each vertical sample.

(b) Construction of the correction matrix for each sample: coefficients are combinations of spectral cross-talk coeffi-
       cients, polarisation cross-talks (provided in CCDB) and lidar constants.

   (c) Computation of the cross-talk corrected signals on the three channels and associated errors.

3. Physical conversion. The purpose of this processing is to retrieve pure range-corrected attenuated backscatter products.
   Lidar absolute constants are applied to retrieve the absolutes signals from spectral cross-talk corrected instrument signals
and associated error products.

## 3.2 CPR

The CPR is a 94 GHz (W-band) cloud radar with Doppler capability that will provide cloud profiles, rain estimates and particle
vertical velocity. Raw instrument data (instrument science packets) as downlinked by the satellite are separated per instrument,
divided into frames of length 1/8 orbit, sorted in time, and stored together with a descriptive product header into level 0 data
products. Level 0 data is delivered from ESA PDGS to JAXA's EarthCARE mission operation system. This is then processed by
CPR level 1b processor which turns the raw data in engineering units into calibrated parameters, such as received echo power
and Doppler velocity, stored in level 1b data products. Geolocations, quality information, and error descriptors are added to the
level 1b products as well.

In CPR level 1b products, received echo power, radar reflectivity factor, normalized surface scattering cross section, Doppler
velocity, spectrum width, and data flag are included.

Gaseous attenuation corrected radar reflectivity factor, unfolded Doppler velocity and various cloud microphysical parame-
ters are not included in level 1b and are processed in level 2a and 2b as shown in Section 5. CPR L1b processor makes the L1b
product for the data observed by Nominal observation mode, Contingency mode, and External Calibration mode. Invalid values
are stored under the effective observation altitude in the L1b product of Nominal observation mode. In the case of External
calibration mode, invalid values are stored in the region higher than 18km. The overview of the CPR L1b processor is described
in the following subsections.





### 3.2.1 Received echo power and Radar reflectivity factor

Received echo power (Pr) is converted from the log detector output of CPR level 0 data. It is integrated with about 500 m horizontal length on orbit. Mean Pr is calculated from division with integration number echo. Mean Pr is calibrated using receiver temperature and calibration load data (hot and normal). Received echo power is distributed before noise power subtraction in consideration of horizontal integration in level 2 processing. The vertical sampling window depends on the observation modes from -1.0 km below the ellipsoid model surface (WGS-84) to 16, 18 or 20 km above the ellipsoid surface for normal observation mode. Radar reflectivity factor (Z) is converted from Received echo power (Pr).

### 3.2.2 Normalized surface scattering cross section (NRCS)

The NRCS is the normalized radar reflectivity which corresponds to the land or ocean surface range. The NRCS is a radar cross section divided by the real cross section. Land and ocean surface range are inferred by surface estimation program written in section 3.2.5.

### 3.2.3 Doppler velocity

Doppler velocities represent vertical movement speeds of echoes if the beam direction of the CPR is precisely nadir. Doppler velocity is derived from IQ detector output instead of log detector output of CPR level 0 data. The echo phase angle $\phi$ is converted from the ratio of the real and imaginary parts of pulse-pair covariance coefficients. A phase change of transmit pulse $\phi_0$ measured from leak signal to CPR receiver during transmit is used for collection of echo phase angle $\phi$. Also, a phase correction $\phi_{sat}$ from satellite speed contamination to the radar beam direction $V_{sat}$ is calculated from ancillary data (satellite velocity, attitude, beam direction). $V_{sat}$ is calculated by dot product of the satellite velocity vector and the unit vector of CPR beam direction.

The Doppler velocity (V) is calculated as follows using $\phi$, pulse repetition frequency PRF and wavelength $\lambda$:

$$V = \frac{\phi \times \lambda \times PRF}{4\pi}$$

The maximum ambiguity Doppler velocity, $V_{max}$, is defined as follows:

$$V_{max} = \frac{\lambda \times PRF}{4\pi}$$

Finally, vertical Doppler velocity from $-V_{max}$ to $+V_{max}$ is derived. Unfolding process and correction of non-uniform reflectivity effect is considered as level 2 processing (Section 5).

### 3.2.4 Surface estimation

Surface echo range is important for the higher level CPR algorithm. In L1b, surface echo range bin is determined using received echo power profile with the help of the digital elevation model information. Also, the exact location of the surface within a



pulse is calculated. In presence of drizzle and/or rainfall, CPR data suffers attenuation and the surface echo may disappear in heavy rainfall cases. This processing produces both the range bin information of the surface and the quality flag of the surface estimation.

## 3.3 MSI

The MSI (Wehr et al., 2023) embarks two cameras; the four channel VNS camera spanning Visible, Near Infrared and two Short Wave IR bandwidths, plus the TIR camera with channels covering three Thermal IR bandwidths. These cameras produce the ISPs that are transmitted to the PDGS and processed to L0 data. MSI L1 processor ingests the MSI L0 product, along with data stored in MSI CCDB, to produce a number of products:

1. **M-NOM** – the nominal L1B data product with radiometrically calibrated imagery and auxiliary support data / quality metrics, including solar irradiance data;

2. **M-RGR** – the nominal L1C data product with radiometrically calibrated imagery, co-registered to TIR band 8, and auxiliary support data / quality metrics, including solar irradiance data;

3. **M-DRK** – containing all calibration products generated from VNS camera dark views;

4. **M-SD1** and **M-SD2** – containing all calibration products generated from VNS camera solar diffuser views (differentiated according to the diffuser used);

5. **M-BBS** – containing all calibration products generated during both the TIR cold-space and TIR calibration black-body views, which includes image statistics, auxiliary parameter statistics and also Failure Detection, Isolation and Recovery (FDIR) status flags;

6. **M-TRF** – containing statistical measurements of ancillary parameters that are involved in TIR sensitivity corrections, which data forms the references against which small deviations are assessed following each TIR CAL used for monitoring for any long-term drifts.

### 3.3.1 Pre-processing

Data from both TIR and VNS channels has already been flat-fielded in the Instrument Control Unit on orbit, via subtraction of an offset that is collected during a regular on-orbit calibration. The flat-field compensates for detector dark current and gain. Initial scrutiny routines in the processor check the validity of the data stream, for instance for continuous error free data and transmission, valid health flags, correct number of channels and sequence counts. Orbital night side data is not expected to contain VNS channels. Data from the TIR camera's three spectral channels must contain at least 19 Ground Lines (GL) of data, which is the Time Delay Integration (TDI) period for which the camera was calibrated on ground and which will be employed on orbit, as well as a channel obtained from reference areas on the 2-D detector. Some supplementary data is contained in a





TIR Auxiliary channel. The processor algorithms require to cover the TDI summing period in order to implement corrections

that use an average over reference area data. Discrete, VNS data channels can be processed independently of each other.

Processing MSI data to valid L1C also requires valid and continuous data over several seconds. The timing data from each GL should differ from the previous one by the GL period of 63 ms. The channel of data that is generated by the Instrument Control Unit (ICU) Auxiliary channel must be valid. If the data passes validity checks, then processing commences via one of three streams, VNS processing, TIR processing or ICU auxiliary processing for the associated house-keeping telemetry.

**3.3.2    VNS processing stream**

VNS data has been corrected on orbit for gross offsets via the flat-fielding that is applied on orbit. Pre-processing first inverts the image contrast, because unprocessed VNS signals decrease as light levels increase. The channel data is subtracted from a display offset that is stored in the CCDB; by default it is the same for all four channels. An average is then calculated over the wing pixel elements for each channel, these being non-illuminated pixels at both sides of each channel's 384 pixel wide,

linear detector array. For each channel, its wing element average signal is subtracted from every pixel sample to compensate for detector dark offset drift and row correlated noise. Thus are generated four streams of VNS data in binary units, each of width 384 pixels (which includes the wing pixels), to which is attached a line specific quality metric and detector_temperature (both from ICU ancillary data), plus the wing_sum (as computed). The four, channel wing-average signals are also made available to the Instrument Calibration and Monitoring Facility (ICMF), for monitoring the health of the VNS detectors, where the average

signal can be correlated against the detector temperature recorded in the ancillary data.

The four streams of VNS data are then radiometrically calibrated into radiance values, with output expressed in $Wm^{-2}$ $\mu m^{-1}$ $sr^{-1}$. Appropriate radiance sensitivity data is selected from the CCDB. Conversion is a linear scaling operation using band-specific and column-specific coefficients from the CCDB that cover all the illuminated columns of VNS data. Each illuminated element in a stream of VNS data is multiplied by the calibration factor from the appropriate calibration factor array. The

conversion to spectral radiance must be performed before any re-sampling or interpolation involved with, for example, re-sampling or re-mapping. The radiometric calibration must also be performed before any of the processing that is associated with a VNS on-orbit calibration (every 16 orbits).

Calibration data is monitored, such that detector degradation and reduction in optical transmission can be compensated. Correction factors for each pixel element in each band (a gain factor), based upon observations collected during the daily VNS

on-orbit calibration, will be updated periodically as part of the ICMF activity. The calibration maintenance gain factors form part of a dedicated parameter set that is utilised by the ECGP.

VNS dark view calibration: This is collected with the aperture shut during the daily VNS calibration and upon any VNS DAY procedure, which occurs following the dark side where VNS imagery is not collected. Radiance statistics are collected, including average signal and temporal noise in each detector element, plus the associated detector temperatures, and form the

calibration product **M-DRK**.

VNS solar view calibration: This is collected at the daily VNS calibration view of the solar illuminated diffuser. The mean solar irradiance and its standard deviation, accumulated for each illuminated column in each VNS band over the entire solar





exposure, as well as the solar SNR, are collected and stored in the calibration product **M-SD1** or **M-SD2**, according to which diffuser was exposed. Data from the CCDB describing the scattering function of the diffuser as a function of detector column, plus the solar illumination angles, are used to derive part of the product.

### 3.3.3 TIR processing stream

The TIR data stream is recovered from the 2-D TIR detector over a 384 column line width. Source channels 5 to 7 are allocated to the three TIR spectral channels, source channel 8 is allocated to reference area data that is summed column-wise from 38 detector reference rows, plus a channel 9 for auxiliary data. The packets received from the three spectral channels have already been subjected to Time Delay Integration (TDI) processing on-orbit (summing over several detector lines), so that each pixel represents a spatial average over the last 19 ground lines. The reference area data has not been subjected to TDI, but is only summed spatially over the 38 detector reference rows; it is used to correct for detector noise and small thermal drifts in the relay lens structure that is viewed by the detector reference areas, over the time period of the TDI processing.

A TIR Display Offset, stored in the CCDB, is first subtracted from each column in the ground line (including reference areas). At each new ground line a column correction must then be applied to the data from each spectral channel, using the reference area data. To this end, at each ground line, the reference channel (which is the sum over 38 detector rows) is stored in a buffer with 19 entries. The reference area temporal average is then computed for each column in the ground line, being the sum of reference area signal in that detector column over the last 19 buffer entries, leading to a vector with 384 samples. The column correction is applied to each TIR band element, subtracting the reference area signal in that column, with scaling for the difference of the single spectral band ground line versus the 38-lines in the reference area.

After column correction an offset correction is applied, associated with the cold space mirror that is used for calibration dark views. In case the ICMF has noticed a degradation associated with instrument sensitivities to certain parameters described below, then sensitivity corrections could also be applied. The signal is then converted to brightness temperature measured in Kelvin, according to band and column specific gain factors stored as look-up table in the CCDB.

TIR blackbody calibration: The gain factors used to convert TIR signal to brightness temperature should result in a Calibration BlackBody (CBB) signal level that is in agreement with the temperature of the CBB that is very accurately measured by precision thermometry during the daily on-orbit calibration. An initial set of gain factors is stored in the CCDB, measured during the ground calibration campaign. The ICMF monitors the TIR transmission in each band and, if needed, generates a new, time-stamped correction factor to be applied to all subsequent processing. The data product from the calibration is **M-BBS** and it is used to generate the TIR gain correction factor.

TIR cold-space view calibration: Cold-space view statistics are collected during each daily, on-orbit calibration and will be monitored at the ICMF. The Cold Space Mirror (CSM) offset correction that is applied to TIR data uses an array of column specific offsets stored in the CCDB. A number of offset files have been generated, based on measurements made during the ground calibration campaign along with calculations that consider a slow degradation of the mirror. Contamination of the cold-space mirror used for the calibration dark view would cause a gradual signal drift. If it is judged to be too large then the drift





can be compensated for by selection of one of the alternative offset parameter files, in order to restore the correct baseline. The data product from the calibration is **M-BBS** and it is used to generate the associated TIR CSM offset factor.

Sensitivity corrections: During the TIR on-ground calibration, the sensitivity of the TIR camera to small perturbations of six parameters was tested, being TIR detector temperature, two TIR bias voltages VFID and VSKIM that relate to gain and offset

settings, TIR relay lens temperature, TIR bench temperature and TIR cover temperature. The calibration involved modifying each of these parameters from the nominal set-point and measuring the camera response to hot and cold targets, in order to measure the offset and gain components at each detector column. Various TIR camera control systems (thermal, electrical, software) should prevent unacceptable deviations of these parameters, however, the processor will flag if any parameter exceeds its limit. Statistical measurements of relevant ancillary telemetry collected during cold space views is stored in the **M-TRF**

data product. The ICMF has the option to impose sensitivity corrections as additional gain and offset parameters, computed using TIR calibration data, however the initial processor configuration does not apply any sensitivity corrections. In case it is required that sensitivity correction(s) is/are imposed, they are applied line by line to the column corrected TIR image data using sensitivity vectors and ancillary parameters from the current line and last TIR calibration.

NB: TIR temperature to radiance and radiance to temperature transforms: Any interpolation operation on TIR data, for

instance for co-registration to L1c, must be applied to radiance values, never to TIR brightness temperature, and the CCDB contains the associated look-up tables. One set of look-up tables contains band specific tables relating the TIR brightness temperature to radiance, for each of the three bands, at a resolution one tenth of the best case NEdT. The band specific inverse transformation tables for radiance to brightness temperature conversions are also supplied. The conversion tables should only be applied to radiometrically calibrated data.

**3.4 BBR**

The BBR is a radiometer that will measure the Total Wave (TW) and reflected Short Wave (SW) radiation from the Earth scene, from which information concerning the emitted Long Wave (LW) is also derived. It collects data in three directions, forward, nadir and aft along track via three different telescopes. A chopper drum with four apertures - two of which house 2 mm thick, curved, quartz filters for filtering of the longwave radiation (i.e to measure the Shortwave only) - rotates continuously around

405 the telescopes, chopping the signal onto the detectors through the sequence SW, drum skin, TW, drum skin etc.

A Calibration Target Drum rotates around the Chopper Drum. The different calibration targets are : hot and cold blackbodies, a sun-diffuser, three fold mirrors and the telescope baffles. the fold-mirrors function is to be able to observe the diffuser from each telescope. Approximately every 80 s the telescopes' views are directed onto a hot or cold black body. SW calibration is performed every two months, by accumulating views of the sun illuminated diffuser over approximately 30 orbits. Monitor

Photo Diodes in the telescope baffles allow aging of the Visible Calibration system (VisCal) to be assessed.

The level 1 processing aims at providing unfiltered radiances over various geographical scales within two products that are elaborated hereunder. However, the user is advised that these products are not directly suitable for science applications, because the instrument effects are not fully removed. Instead, the user is advised to use the (so-called unfiltered) product BM-RAD:





1. **B-NOM** – the nominal BBR products averaged over various geographical scales:

   – standard product: averaged over 10 km along-track and 10 km across-track using a trapezoidal weighting function of the Point Distribution Function (PDF).

   – small product: averaged over 10 km along-track and 5 km across-track using a rectangular weighting function of the PDF.

   – full product: averaged over 10 km along-track and over the full range of the product across-track.

2. **B-SNG** – the single pixel product in which the filtered radiance is provided at the pixel-resolution.

The ECGP algorithm has three processing steps. Step 1 operates on the incoming ISP (Instrument Source Packets), encapsulated as the L0 product. Step 2 annotates the incoming per-telescope data with the correct gains and offsets to apply derived from the calibration sequences. Two ageing (calibration) products are output from this stage:

1. **B-SOL** – the solar calibration BBR L1 product. The BBR Solar Calibration data product is a collection of the SW ageing views over time.

2. **B-LIN** - the linear calibration BBR L1 product. The BBR Linear Calibration data product is a collection of the measured gain (and offset) while viewing the blackbodies (BBs) during a linearity check.

Step 3 takes each data stream (one per telescope) and performs the radiometric corrections and the various spatial summations to generate the measurement level 1 products: B-NOM and B-SNG. These three steps are described further hereafter.

### 3.4.1 Marshalling stage

The processed ISPs from BBR are arranged in the L0 chronologically with 24 acquisitions (8 per telescope), per ISP. This stage takes the L0 product and rearranges the data as an acquisition stream per telescope. At this stage, the relevant house-keeping (HK) and satellite information is annotated into the data. The sequence is:

– Output an intermediate file per ISP containing the measured resistance temperatures (using current/temperature Look Up Tables) per telescope, along with black-bodies and instrument temperatures.

– Quality check: check if a corrupted flag has been raised or if data are insufficient within one ISP (reject if so). An additional flag is potentially raised per pixel based on the generated voltage quality check (to highlight high contrast scenes at pixel level).

– Annotate the ISP with the exposure type: the calibration drum position, to designate Total Wave, Short Wave or a View of the drum inner surface (if the position differs for two successive acquisitions then the drum is considered as moving for the first acquisition which is flagged as invalid).

– Annotate the ISP with the spacecraft position and pointing observation.



- Generate an intermediate file of telescope (i.e. three files) data based on the incoming values.

- Generate an intermediate file of the output values of the monitoring photodiodes, when relevant (i.e when a visible calibration is performed) along with the concerned telescope.

### 3.4.2 Radiometric calibration

At this stage, the incoming per-telescope data are annotated with the gains and offsets to apply, derived from the calibration sequences. The geo-location information are also added. Preliminary radiometric corrections are performed and the two calibration products are generated at this stage.

- Chopper subtraction: the background radiation generated by the chopper drum is subtracted from the voltage measured for each acquisition.

- Geolocation: The first operation is to create the ground positions (along-track) representing the pixel barycentres. This is derived from the spacecraft attitude data by computing the intersection of the line of sight using the EOCFI dedicated routines. Secondly the ground coordinates are derived for each of the measurement pixels. Finally these are also translated into along-track and across-track coordinates with respect to the barycentre ground-track. All these information are annotated to the telescope files.

- Gain calculation: this calibration processing is performed on a telescope-by-telescope basis. It consists of determining the gains and offsets for the TW and SW channels for each telescope. For a given calibration sequence, typically four measurements (two TW and two SW) of each internal blackbodies (hot and cold) are performed. After averaging over the available measurements in the sequence, two values are derived, being, the voltages TW and SW for each blackbody (these values are corrected from the dark current). The radiance of each black body is then interpolated in a two-dimensional look-up table (dependent on blackbody and telescope temperatures) using the intermediate file created during the marshalling stage. Offsets for SW and TW channels are directly given by the voltages measured for the cold blackbody. The TW gain is given by the ratio of the difference of voltage for hot and cold BB and the difference of radiance for the two BBs. Finally, the SW gain is obtained by multiplication of the TW gain by the SW Gain Ratio (value is extracted from the CCDB). This ratio is first characterized on ground but is affected by the ageing of the quartz filter. For this reason it is regularly updated during the solar measurement (VisCal). These calibration values are stored in the B-LIN which is produced at this stage.

- Output B-SOL: this stage parses the telescope file to generate the calibration file corresponding to the SW calibration position (once per orbit) which contains acquisition time, sun position, signal levels on the three telescopes (both when the SW filter is in place and not), gains and offsets information, as well as quality information. This data is used, following careful inspection by the ICMF, to assess if there is any ageing of the SW spectral response. If this this is the case, the SW Gain Ratio parameter will be modified accordingly.





### 3.4.3 Integrate and create products

For the B-NOM product, the following processing sequence is applied: first a list of product barycentre is produced at a 1 km interval (the processor determines what the barycentres location should be with respect to the along-track pixel positions). Then for the three different resolutions of the product (standard, small, full) it is determined from the along-track dimension, which pixels are to be integrated in the product. Conversion of along-track/across-track into geodetic latitude/longitude is also performed. Depending on the associated weighting function (dependent on the product resolution), a weight is then associated
to each individual pixel. After correction of the offset and the gain for each channel (TW and SW), the integration of the weighted product is performed over the pre-determined area for each resolution. The LW radiances are then derived from the subtraction of the SW radiances from the TW ones, keeping in mind the differential gain of the channels.

For the B-SNG product, the gain and offset corrected radiances are provided at pixel level and for the TW and SW channels (no LW radiances are derived).

## 4 Auxiliary data (level 1d) processors and products

### 4.1 X-MET: Meteorological fields for the EarthCARE swath

The X-MET processor generates the EarthCARE meteorological product, X-MET, using meteorological fields selected from output of the ECMWF Early Delivery System of the Atmospheric Model high-resolution 10-day forecast (HRES) model and sub-setting them to the EarthCARE swath while keeping them on the original model grid. This reduces significantly the
data volume as compared to using global meteorological fields (reduction by a factor 20) or as compared to interpolation to EarthCARE instrument grids or the EarthCARE Joint Standard Grid.

The X-MET product format is compliant with the generic EarthCARE product format and EarthCARE metadata conventions. It includes certain derived parameters, being tropopause height (both for the World Meteorological Organisation (WMO) and CALIPSO definitions) and wet bulb temperature. The X-MET swath is based on the EarthCARE ideal orbit and includes
an allowance for the dead band, within which the actual orbit is maintained.

X-MET products are generated four times per day, according to the production schedule of ECMWF Early Delivery data by its Integrated Forecast System (IFS). Each production run covers about 20 hourly forecasts, which, considering EarthCARE 1/8 orbit granularity, corresponds to 104 X-MET products per run. This generally results in multiple X-MET products being available for any selected time.

Horizontal grid: Parameters are provided on ECMWF model grid points, which is a reduced Gaussian grid. ECMWF uses an octahedral grid, with a resolution between 8 and 10 km, and a total number of 6.59 million grid points. However, only grid points within a 330 km swath around the EarthCARE ground track are included in X-MET, leading to about 26,900 grid points per frame.

Vertical grid: Pressure and geometric altitude on model levels are provided in X-MET for each point of the horizontal grid,
which enables interpolation/conversion from model levels to pressure or altitude coordinates. Unlike other EarthCARE data





products, which have the lowest altitude level at or below the Earth reference ellipsoid independent of the topography, for X-MET, lowest altitude level is the actual surface level, i.e. it follows the topography.

Should the ECMWF model resolution change in the future, X-MET will follow, i.e., it will always be provided at the original model resolution.

Grid point coordinates (latitude, longitude, altitude) are provided within the X-MET product. Parameters in X-MET are interpolated linearly between two adjacent forecasts (or the analysis and the first forecast) to a representative EarthCARE sampling time (mid-time of the frame). The temporal sampling of the forecasts between which this interpolation takes place is one hour. As X-MET contains data interpolated in time to the EarthCARE sampling time, processors using X-MET do not need to interpolate in time, only in space. Relevant times are reported in the Specific Product Header.

The X-MET processor runs inside a Kubernetes cluster on a platform provided at ECMWF, with access to the output from their HRES model. It will be managed and monitored by ECMWF, whilst the pods - containers with shared storage and network resources, used to transfer the X-MET product - will be managed by the PDGS.

## 4.2   X-JSG: The Joint Standard Grid

This processor creates the EarthCARE Joint Standard Grid product, X-JSG, from the geolocation in ATLID and CPR level 1b
products. X-JSG contains the common spatial grid used across instruments in EarthCARE level 2 processing, called Joint Standard Grid, and a number of additional geolocation parameters widely used in level 2 processing; a land flag, terrain elevation, solar angles, viewing angles and some index and count parameters. X-JSG itself does not contain any instrument observation data. It ensures that the data from radar, lidar and multispectral imager can be collocated such that they are observing the same column of the atmosphere. The X-JSG level 1d grid product consists of two, two-dimensional (2D) grids, creating a three
dimensional grid that follows the EarthCARE track. X-JSG is constructed by combining:

 – The CPR horizontal grid along track (combining two CPR pixels for every JSG pixel, or approximately 1 km),

 – The ATLID vertical grid, and

 – A fixed 1 km grid to extend the grid across track to cover the complete MSI swath.

In case of non-availability of either CPR or ATLID measurement data, the X-JSG processor falls back to a contingency
solution. This is implemented, for instance, during instrument calibration modes. In such instances the grid will be constructed without the respective instrument, using a similar but fixed sampling. In case of non-availability of both CPR and ATLID measurement data for a complete frame, no JSG will be produced. However, in such instances synergistic retrievals, without measurements from the active sensors, are not required. The horizontal grid point locations define the X-JSG pixel centres. Pixels across track, for a given along-track position, share the same vertical grid. The vertical grid is specified for each grid
point along track, as the position of the vertical grid points varies along track due to the dynamic adaptation of the ATLID range to the satellite height, combined with a finite step size for the range.

Along track: The spatial extent of the along-track grid is equal to a frame (1/8 orbit), which is the granularity of all Earth-CARE products. However, the CPR creates profile groups of 14 CPR profiles, between which there is a larger spacing, which





results in a series of seven X-JSG pixels. The nominal X-JSG grid is irregular due to this way of CPR sampling. In case of no
CPR measurement data, X-JSG will use a fixed 1000 m horizontal grid point distance (configurable).

Across track: Set via a configuration that defines a fixed 1 km grid sampling and the left and a right boundary dimensions, selected based on MSI sampling and swath size.

Vertical: Vertical grid point distance is defined by ATLID vertical sampling, which is about 103 m up to an altitude of 20 km and 500 m above this altitude. The lower and upper boundaries of the vertical grid are set via configuration parameters. In case
of no ATLID measurement data, a vertical grid is created with two possible vertical grid point sample distances, high-resolution at 100 m sample distances and low-resolution at 500 m vertical sample distances. The high and low resolution sample distances, as well as the altitude boundary of these sampling regions, are configurable.

This product facilitates the application of synergistic (L2b) classification and retrieval algorithms as well as the synergistic use of a number of single-instrument (L2a) products. It is closely linked to the merged observations product (AM-MO) which
directly uses the JSG definition in order to produce a file containing co-located observations.

## 5    Level 2 data processors and products

EarthCARE level 2 data products include a comprehensive range of geophysical parameters related to aerosols, clouds, precipitation and radiation. Wehr et al. (2023) (section 8.3) give an overview of the ESA and JAXA L2a/L2b data products containing retrieved aerosol, cloud, precipitation and radiation parameters and supporting science activities. In this section we provide
brief descriptions of all level 2 data products and references to corresponding publications which contain detailed information about the products, retrieval algorithms and their verification and validation. Tab. 1 and 2 lists all ESA and JAXA level 2a (L2a) data products, i.e., data products derived from one EarthCARE instrument only, along with the reference for their detailed description. Data products synergistically retrieved from more than one EarthCARE instrument are referred to as level 2b (L2b) data products. Inputs to L2b processors may be L1b/c/d data products as well as L2a and other L2b data products. Tab. 3
and 4 lists all ESA and JAXA L2b data products including their respective reference. All products listed in Tab. 1 and 3 are operationally produced by the ESA PDGS. For products listed in Tab. 2 and 4, Standard Products are operationally produced by the JAXA Satellite Applications and Operations Center (SAOC), while Research Products are products by JAXA Earth Observation Research Center (EORC) and/or Japanese institutes/universities. All products will be made available to users usually not later than 48 hours after sensing.

While not part of the ESA or JAXA data products, the development of a cloud climate product processor for ATLID is relevant in this context. It is designed with the purpose of making ATLID cloud observations directly available for climate applications (Feofilov et al., 2023).





## 5.1 Single-instrument (level 2a) processors and products

### 5.1.1 ESA single-instrument processors and products

ESA data products generated from measurements of a single instrument are listed in Table 1. There is no BBR level 2a product, as scene information from the MSI is required to derive radiances and fluxes from BBR. Level 2a products are usually provided at instrument resolution or multiples thereof, ATLID and CPR on a vertical "curtain" (dimensions: along track and altitude), and MSI on a horizontal "swath" (dimensions: along track and across track).

For each of the active instruments, ATLID and CPR, two classification products are produced, a feature detection mask that
provides areas of significant return (A-FM and C-FMR) and a target classification that identifies various classes of hydrometeors and aerosols (A-TC and C-TC). From the radar reflectivity corrected for gaseous attenuation and non-uniform beam filling (also in C-FMR), and Doppler velocities corrected for antenna mispointing, non-uniform beam filling and velocity folding (C-CD), are derived vertical profiles of cloud and precipitation microphysical parameters (C-CLD). The primary ATLID level 2a product contains vertical profiles of extinction, backscatter and depolarisation (A-EBD). This is the starting point for further
ATLID products covering aerosol profiles (A-AER), ice cloud profiles (A-ICE) and layer information for clouds (A-CTH) and aerosols (A-ALD).

Level 2 processsing for MSI starts with a classification product as well (M-CM). This contains the cloud flag, cloud phase and cloud type. In a second stage, cloud optical and physical properties, namely cloud optical thickness, effective radius and top height are derived (M-COP), as well as aerosol optical thickness (M-AOT).

### 5.1.2 JAXA single-instrument processors and products


Table 2 shows all JAXA L2a products and their product category shown in the JAXA Production Model in Figure 2. Short descriptions of the JAXA L2a products are as follows:

CPR_ECO – containing 1 km and 10 km horizontally integrated radar reflectivity and Doppler velocity. Clutter echo correction and gas attenuation correction are provided for radar reflectivity, and Doppler unfolding correction is provided for the
Doppler velocity.

CPR_CLP – containing cloud mask, cloud particle type and cloud microphysics, which are derived mainly from radar reflectivity factor.

ATL_CLA – providing classification for each ATLID observation grid and optical properties of cloud and aerosol such as extinction coefficient, backscatter coefficient, lidar ratio, and depolarisation ratio. Planetary boundary layer height is also
contained.

MSI_CLP – cloud product derived from MSI, including cloud flag and phase information and water cloud properties such as effective radius and optical thickness. Furthermore, cloud top temperature, pressure and height are provided. Ice cloud properties are in MSI_ICE product.

CPR_DOP – the corrected Doppler velocity, considering inhomogeneity and unfolding is included, in addition to the correction for Doppler velocity performed in CPR_ECO.



| Product Name | *Processor Name* | Derived from instrument | Content | Reference |
|---|---|---|---|---|
| C-FMR | *C-PRO* | CPR | Feature mask and corrected reflectivity | Kollias et al. (2023) |
| C-CD | | | Corrected Doppler | Kollias et al. (2023) |
| C-TC | | | Target Classification | Irbah et al. (2023) |
| C-CLD | *C-CLD* | CPR | Cloud profiles | Mroz et al. (2023) |
| A-FM | *A-FM* | ATLID | Feature mask | van Zadelhoff et al. (2023) |
| A-AER | *A-PRO* | ATLID | Aerosol profiles | Donovan et al. (2023b) |
| A-ICE | | | Ice water content and effective radius | Donovan et al. (2023b) |
| A-TC | | | Target classification | Irbah et al. (2023), Donovan et al. (2023b) |
| A-EBD | | | Extinction, backscatter, depolarisation | Donovan et al. (2023b) |
| A-CTH | *A-LAY* | ATLID | Cloud top height | Wandinger et al. (2023) |
| A-ALD | | | Aerosol layer descriptor | Wandinger et al. (2023) |
| M-CM | *M-CLD* | MSI | Cloud mask and phase | Hünerbein et al. (2023b) |
| M-COP | | | Cloud optical and physical properties | Hünerbein et al. (2023a) |
| M-AOT | *M-AOT* | MSI | Aerosol optical thickness | Docter et al. (2023) |

**Table 1.** References of all ESA L2a products and their retrieval algorithms shown in the ESA Production Model, Figure 1.

CPR_RAS – providing parameters related to rain and snow, such as rain water content, snow water content, ratio rate, snow rate. Attenuation corrected radar reflectivity is also contained. Two types of rain water content and snow water content, with and without Doppler velocity, are available.

CPR_VVL – containing vertical air motion in cloud regions and sedimentation velocity of cloud particles.

ATL_ARL – extinction coefficients for four aerosol components (dust, black carbon, sea-salt and water-soluble particles) are included.

MSI_ICE – including optical thickness, effective radius and cloud top temperature/pressure/height of ice cloud. Water cloud properties are in MSI_CLP product.

MSI_ARL – aerosol optical thickness and Angstrom exponent derived from MSI are contained. Angstrom exponent is avail-

able only over ocean.





| Product Name | Product Category | Derived from instrument | Content | Reference |
|---|---|---|---|---|
| CPR_ECO | Standard | CPR | Radar reflectivity and Doppler velocity | Hagihara et al. (2021) |
| CPR_CLP | Standard | CPR | Cloud mask/type and cloud optical properties | Hagihara et al. (2010), Kikuchi et al. (2017), Okamoto et al. (2010), Sato and Okamoto (2011), Okamoto (2023) |
| ATL_CLA | Standard | ATLID | Feature mask, target mask, and cloud/aerosol optical properties | Nishizawa et al. (2019); Nishizawa (2023); Hagihara et al. (2010); Yoshida et al. (2010); Oikawa (2023) |
| MSI_CLP | Standard | MSI | Cloud flag/phase and water cloud properties | Nakajima et al. (2019), Wang et al. (2023) |
| CPR_DOP | Research | CPR | Doppler velocity correction value and unfolding value | Hagihara et al. (2021); Hagihara (2023) |
| CPR_RAS | Research | CPR | Rain/Snow water content, Rain/Snow rate | Sato et al. (2009); Sato (2023) |
| CPR_VVL | Research | CPR | Vertical air motion and sedimentation velocity | Sato et al. (2009); Sato (2023) |
| ATL_ARL | Research | ATLID | Aerosol extinction coefficient of water soluble, dust, sea salt, and black carbon | Nishizawa et al. (2008, 2011) |
| MSI_ICE | Research | MSI | Ice cloud properties | Letu et al. (2016, 2018) |
| MSI_ARL | Research | MSI | Aerosol optical thickness and Angstrom exponent | Yoshida et al. (2018) |

**Table 2.** References of all JAXA L2a products and their product category shown in the JAXA Production Model, Figure 2.

## 5.2 Synergy (level 2b) processors and products

### 5.2.1 ESA synergy processors and products

ESA data products generated from measurements of two or more EarthCARE instruments are listed in Table 3. Level 2b
products are usually provided on the Joint Standard Grid described in section 4.2.



The single-instrument target classifications (A-TC and C-TC) are used to derive a synergistic target classification (AC-TC). Synergistic ATLID/MSI layer products are generated for clouds (AM-CTH) and aerosols (AM-ACD). The primary synergy product for cloud, aerosol and precipitation parameters is ACM-CAP, which uses an optimal estimation scheme. In addition, and as a backup for ACM-CAP, a simpler synergy product ACM-COM contains a best estimate of cloud and aerosol profiles based on a composite of level 2a products. As a preparation to the radiative transfer calculations, a three-dimensional scene is constructed finding nadir pixels that match off-nadir pixels in MSI radiance (ACMB-3D). In this way, the cloud, aerosol and precipitation fields from ACM-CAP (or as a backup, from ACM-COM) can be extended out to 15 km across-track and used in 1D and 3D radiative transfer models to derive broadband radiances, fluxes and heating rate profiles (ACM-RT). Broadband radiances (BM-RAD) and fluxes (BMA-FLX) are also derived directly from BBR measurements. In a final assessment step (ACMB-DF), broad-band radiances and fluxes from radiative transfer models are compared to the ones from BBR.

### 5.2.2   JAXA synergy processors and products

Table 4 shows all JAXA L2b products and their product category shown in the JAXA Production Model in Figure 2.

Short descriptions of the JAXA L2b products are as follows:

AC_CLP – similar parameters as CPR_CLP are estimated using CPR-ATLID, however, the number of grids with valid values increases compared with CPR_CLP because AC_CLP is derived from both CPR and ATLID, which have different sensitivity for thin and deep clouds.

ACM_CLP – similar parameters as CPR_CLP and ACM_CLP are estimated using CPR-ATLID-MSI. In addition, liquid water path and ice water path are available in ACM_CLP.

ALL_RAD – containing radiative fluxes and heating rate for both shortwave and longwave regions derived with 1D plane-parallel radiative transfer simulation.

AC_MRA – containing two types of mass ratio of 2D ice to ice water content. One is derived with Doppler velocity and the other without Doppler velocity.

AC_RAS – providing rain water content, snow water content, rain rate and snow rate, derived utilising both CPR and ATLID data. Two types of rain water content and snow water content, with and without Doppler velocity, are available.

AC_VVL – providing information related with atmospheric vertical motion estimated with CPR and ATLID data. The parameters of vertical air motion and sedimentation velocity are included.

AM_ARL – providing extinction coefficient of each aerosol type (dust, black carbon, sea-salt, and water-soluble particles) and mode radius for fine mode and coarse mode. Aerosol direct radiative forcing at Top Of Atmosphere (TOA) and Bottom Of Atmosphere (BOA) are included as well.

ACM_CDP – containing cloud mask, cloud particle type, liquid/ice water path and cloud microphysics derived from ATLID, CPR and MSI data. ACM_CDP utilizes Doppler velocity.

ACM_RAS – containing same parameters as AC_RAS, but ACM_RAS is derived from three sensors of ATLID, CPR and MSI.



| Product Name | Processor Name | Derived from instruments | Content | Reference |
|---|---|---|---|---|
| AM-CTH | *AM-COL* | ATLID, MSI | Cloud top height | Haarig et al. (2023) |
| AM-ACD | | | Aerosol column descriptor | Haarig et al. (2023) |
| AM-MO | *AM-MO* | ATLID, MSI | ATLID & MSI level 1b merged onto same grid | |
| BM-RAD | *BM-RAD* | BBR, MSI | Broad-band radiances (unfiltered) | Velázquez Blázquez et al. (2023a) |
| BMA-FLX | *BMA-FLX* | BBR, MSI, ATLID | Broad-band fluxes | Velázquez Blázquez et al. (2023b) |
| AC-TC | *AC-TC* | ATLID, CPR | Synergistic target classification | Irbah et al. (2023) |
| ACM-CAP | *ACM-CAP* | ATLID, CPR, MSI | Cloud, aerosol, precipitation best estimates | Mason et al. (2023b) |
| ACM-COM | *ACM-COM* | ATLID, CPR, MSI | Composite cloud and aerosol profiles | Cole et al. (2022) |
| ACMB-3D | *ACMB-3D* | ATLID, CPR, MSI, BBR | Constructed three-dimensional scene | Qu et al. (2023) |
| ACM-RT | *ACM-RT* | ATLID, CPR, MSI | Radiative transfer products - fluxes, radiances, heating rates from 1D and 3D models applied to retrieved cloud/aerosol/precip scenes | Cole et al. (2022) |
| ACMB-DF | *ACMB-DF* | ATLID, CPR, MSI, BBR | Differences between radiances and fluxes calculated from retrievals (ACM-RT) and BBR measurements (BM-RAD, BMA-FLX) | Barker et al. (2023) |

**Table 3.** References of all ESA L2b products and their retrieval algorithms shown in the ESA Production Model, Figure 1.

ACM_VVL – containing same parameters as AC_VVL, but ACM_VVL is derived from three sensors of ATLID, CPR and

MSI.

ACM_ICE – containing effective radius and optical thickness of ice cloud derived with the emission method called MWP method (Multi-wavelength and multi-pixel method).



| Product Name | Product category | Derived from instruments | Content | Reference |
|---|---|---|---|---|
| AC_CLP | Standard | ATLID, CPR | Cloud mask/type and cloud optical properties | Okamoto et al. (2007, 2008, 2010), Hagihara et al. (2010), Kikuchi et al. (2017), Sato and Okamoto (2011) |
| ACM_CLP | Standard | ATLID, CPR, MSI | Cloud mask/type and cloud optical properties | Okamoto (2023) |
| ALL_RAD | Standard | ATLID, CPR, MSI, BBR | SW/LW radiative flux and radiative heating rate | Oikawa et al. (2013, 2018), Okata et al. (2017), Yamauchi (2023) |
| AC_MRA | Research | ATLID, CPR | Mass ratio (2D Ice/IWC) | Sato et al. (2009); Sato (2023) |
| AC_RAS | Research | ATLID, CPR | Rain/snow water content and rain/snow rate | Sato (2023) |
| AC_VVL | Research | ATLID, CPR | Vertical air motion and sedimentation velocity | Sato et al. (2009); Sato (2023) |
| AM_ARL | Research | ATLID, MSI | Aerosol extinction coefficient of water soluble, dust, sea salt, and black carbon, mode radius | Kudo et al. (2016) |
| AM_ARL | Research | ATLID, MSI | Aerosol direct radiative forcing | Oikawa et al. (2013, 2018) |
| ACM_CDP | Research | ATLID, CPR, MSI | Cloud mask/type and cloud optical properties with Doppler velocity | Okamoto (2023) |
| ACM_RAS | Research | ATLID, CPR, MSI | Rain/snow water content and rain/snow rate | Okamoto (2023) |
| ACM_VVL | Research | ATLID, CPR, MSI | Vertical air motion, sedimentation velocity | Okamoto (2023) |
| ACM_ICE | Research | ATLID, CPR, MSI | Ice cloud optical properties with emission method | Okamoto (2023) |

**Table 4.** References of all JAXA L2b products and their retrieval algorithms shown in the JAXA Production Model, Figure 2.





# 6 Development of the EarthCARE processing chain

## 6.1 End-to-end simulator E3SIM

Development and test of the processors was made possible using a dedicated EarthCARE End-to-End Simulator (E3SIM). Test data, consisting of input scene files that describe the physical properties, were assembled by assimilation of data from many sources. Forward models were used to operate on the input scene files and generate the expected input to each of the instruments, with instrument models to simulate the engineering data that would be output from the instruments and transmitted to the ground. The L0, L1 and L2 processors could then be used to generate the corresponding data products, which need to be

compared against the input data. The E3SIM is optimised for producing small amounts of highly representative data to be used in algorithm development. It is not meant to be used for routine operational generation of data products.

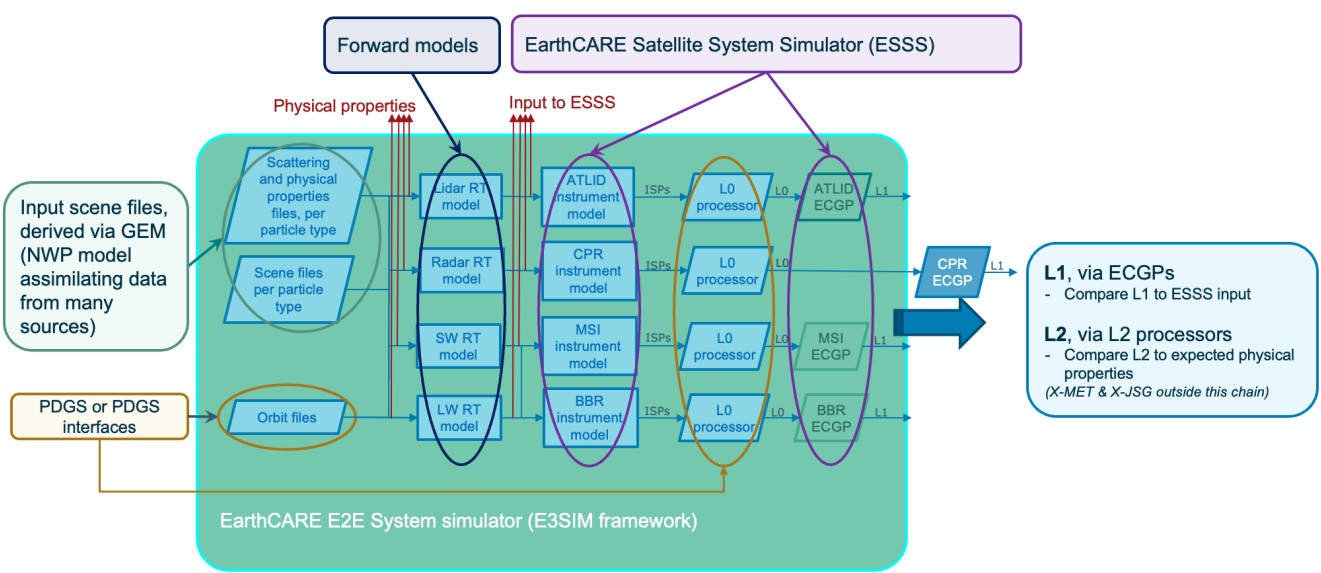

**Figure 3.** Simulation chain developed for the EarthCARE processors development. RT=Radiative Transfer, SW=Short Wave, LW=Long Wave, NWP=Numerical Weather Prediction, E2E=End to End, ECGP=EarthCARE Ground Processor

E3SIM consists of the simulator framework, which allows to configure and run simulations of the complete processing chain or any part of the chain (down to individual processor runs), and the individual processor modules that are plugged into the simulator. Processors are connected to each other via their products and outputs of one processor in the chain are inputs to one

or more processors further down in the chain. The production model (Figure 1) is implemented via "task tables", which describe the inter-dependencies between processors in a general way. From these, "job orders" are generated as the basis for specific production runs, listing all required input data for a given run. This interface is the same as the one used by the operational PDGS, simplifying the development and testing as the processors have to support only a single interface.





## 6.2 Test data sets

The E3SIM was used to generate a number of test data sets for the processor development. Test data for level 1 processor verification used scenarios from the EarthCARE system requirements wherever possible. These typically cover the extremes of dynamic ranges, from detection limit to saturation. Level 2 processors used a wide range of input scenes, from simple synthetic scenes in the early stages of development, via scenes based on campaign data and satellite measurements, specifically the A-train instruments CloudSat, CALIPSO and MODIS, to the three science reference frames, Halifax, Baja and Hawaii,

based on the Canadian numerical weather prediction model, GEM. This dataset and its generation are described in detail by Qu et al. (2022) and Donovan et al. (2023a). It covers a wide range of cloud and aerosol scenes and was used extensively in the development, verification, and inter-comparison (Mason et al., 2023a) of the EarthCARE level 2 processors.

A dataset from a global storm-resolving simulation, and its use in the global evaluation of CPR Doppler velocity errors, is described by Roh et al. (2023) and Hagihara et al. (2023).

## 6.3 Collaborative development


The large number of data processors and processor developers, and the complex production model with its many inter-dependencies, required an efficient setup for enabling collaboration between developers, sharing code and information, testing and managing software problems. We used a highly integrated, collaborative software development environment, based on a mix of proprietary and open source tools such as Atlassian Confluence for sharing information and minutes of meeting, Atlas-

sian JIRA for reporting software problems and planning work, gitea as a code repository, Jenkins for running automated tests, and MinIO as a test data repository.

## 6.4 Lessons learnt

After more than 15 years of developing the EarthCARE processing system, it is useful to look back and reflect on what could have been done better and what worked well.

We found that it is important to systematically build up the end-to-end simulation chain and keep it consistent along the way. Interfaces and reference software environments should be defined and documented very early. It is useful not to develop everything in parallel from the start, instead to build a single chain first (preferably the one for the "simplest" instrument), then review what can be improved, feed that into general requirements, and apply the lessons learned to remaining chains (which can then be done in parallel). Level 1 processors need to be connected to the level 2 processing chain early.

Consistency of the simulation and processing chain (including input data) should be verified regularly. Late (fundamental) changes in the processing chain must be avoided. If they do happen, they must include a corresponding adaptation of other parts of the chain to keep it consistent (example: level 1 processor update might require an instrument simulator update).

Expertise and developments should be organised "vertically" along instrument processing chains (e.g. ATLID, CPR), rather than "horizontally" along processing levels (e.g. instrument simulator, level 1 processor).





Test data sets should be systematically built up and curated in a dedicated and not to be underestimated effort along the development. Rigorous configuration control is required for test data sets (including the CCDB), making them fully traceable. Test data set limitations should be documented at every development step. Test data should contain (i) synthetic, simple test data sets allowing simple verification along the development process, (ii) data sets using actual instrument data (in particular for internal calibration modes), (iii) state-of-the art scientific data sets, (iv) long data sets for GS verification.

Master repositories should be used for code, test data and documentation. The same item should not be stored in multiple places (other than synchronised repositories for backup).

Level 2 processor developers should be involved early in the L1 algorithm/product verification, with bread-boarding of critical modules. Development contracts should be short (not longer than 2.5 years) to keep flexibility. Agile developments are to be preferred, they worked very well for EarthCARE. All delivered code needs to be used immediately, regularly, and by as

many people as possible, so issues can be identified and fixed early, while software developers are still familiar with the code.

We found that it worked well to have a set of general requirements and conventions across all processors and to limit allowed programming languages to the absolute minimum (C/C++/Fortran); this simplified the maintenance. Strict processor runtime requirements turned out to be very useful, forcing developers to optimise runtime. We had an efficient set-up of software support to the scientific level 2 processor developers, and the collaborative environment described in the previous section

helped to coordinate the development in an efficient way.

## 7   Product format and conventions

### 7.1   Product format

EarthCARE level 1 and level 2 data products will be provided as NetCDF-4/HDF5 files. This format has been selected as it is widely used in the community and is self-describing. Furthermore, a large number of software tools and libraries are available

to read this format. Products use internal data compression and each product covers one frame (1/8 orbit or about 5000 km along track).

### 7.2   Product conventions

For ESA products and processors, each data product name and each processor name consists of two parts, up to four letters XXXX, indicating the data origin and up to three letters YYY indicating data content, connected by a hyphen: XXX-YYY.

For most products, XXX refers to one or more of the four EarthCARE instrument (A for ATLID, C for CPR, M for MSI, B for BBR). For example, the level 2a product A-FM refers to the feature mask (FM) derived from ATLID (A). Auxiliary data products X-MET and X-JSG (4) do not use any instrument measurements, so they use X instead of an instrument identifier.

JAXA data products are referenced by their product identifiers that are composed of two parts separated by an underscore (e.g. CPR_CLP). The first three letters indicate the instruments used in the product (CPR=CPR, ATL=ATLID, MSI=MSI,





AC=ATLID-CPR, ACM=ATLID-CPR-MSI, AM=ATLID-MSI, ALL=Four sensors) and the latter three letters indicates the geophysical content of the product (e.g. CLP for cloud properties and RAD for radiation).

Parameters and dimensions within the data products follow a common set of conventions, which aim to harmonise naming across products and to make the names as self-explanatory as possible. This means they can be rather long, as acronyms (apart from the instrument names) are avoided. Other conventions concern the use of units and of certain standard suffices, such as

"flag" for a parameter that can only assume the values 0 and 1.

## 8    Conclusions

We have presented an overview of the EarthCARE processors development, encompassing processors developed by teams in Europe, Japan and Canada. These will facilitate exploitation of the four instruments embarked on the EarthCARE satellite, which will fly in a sun-synchronous, low Earth orbit with a Mean Local Solar Time of 14:00 ±5 minutes. As well as the

single instrument products available at level 1 and level 2, a comprehensive set of multiple-instrument, synergistic, level 2 data products will retrieve aerosol, cloud, precipitation and radiation parameters. A Joint Standard Grid is generated and used to ensure that radar, lidar and multispectral image data is collocated such that the same column of the atmosphere is viewed. An auxiliary product contains a sub-set of meteorological fields from an ECMWF atmospheric model that covers the EarthCARE swath.

Test of the processing chain has made use of a dedicated End-to-End Simulator, called E3SIM, and special test data sets that were generated from data assimilated from multiple sources. E3SIM incorporates forward models and instrument simulators to generate the signals expected in Instrument Source Packets. The ISPs are ingested into the processor chain and outputs can be compared against the input data. A description of the collaborative development used in the processing chain development is presented as well as the lessons learnt.

All EarthCARE data products will be available from both ESA and JAXA website. An overview is given of EarthCARE product format and conventions, as well as a short summary of the different products' contents and references to the dedicated papers that describe their algorithm and content in more detail.

EarthCARE will allow scientists to evaluate the representation of cloud, aerosol, precipitation and radiative flux in weather forecast and climate models, with the objective to improve parameterisation and, in particular, address uncertainty in cloud

processes.

*Code and data availability.*   This overview paper refers generally to software code for level 2 processors and for atmospheric models that have been used to prepare data and simulations. The references for software repositories can be found in each of the dedicated papers that are contained within this special issue on EarthCARE level 2 algorithms and data products. The EarthCARE level 2 demonstration products from simulated scenes are available at https://doi.org/10.5281/zenodo.7117115 (van Zadelhoff et al., 2023b).



*Author contributions.*   ME prepared the manuscript and overview of the processing chain and level 2 processors, simulation and test. FM described the ATLID and BBR instrument and L1 processors. KW introduced the mission and described the MSI instrument and L1 processor, and the auxiliary processors. TK described the CPR instrument and L1 processor, with contributions from NT and YO. TT described the JAXA L2 processing chain. TW contributed to the science background and data product overview. DB, the ESA EarthCARE Project Manager, and ET, the JAXA Project Manager, provided overall guidance on the technical and programmatic context.

*Competing interests.*   The authors declare that they have no conflict of interest.

ssue statement. This article is part of the special issue "EarthCARE level 2 algorithms and data products". It is not associated with a conference.

*Acknowledgements.*   We would like to recognise the industrial partners working on the satellite, payload and level 1 processors, led by
the satellite prime, Airbus Defence and Space (Germany), with instrument teams at NEC (Japan), SSTL (UK), Airbus Defence and Space (France), Thales Alenia Space (UK), and level 1 processors development and auxiliary processors development supported by GMV (UK) and S[&]T (NO) respectively. We thank the teams of scientists at several institutions in Europe, Japan and Canada that have developed EarthCARE level 2 processors and products, as well as at the ECMWF. Special thanks to members of the Joint Mission Advisory Group, which advises ESA and JAXA on science aspects of the mission, as well as to the principal investigators and project scientists of NASA
missions CloudSat and CALIPSO and the CERES instrument for their long-standing support and advice. We would also like to thank colleagues at ESA, JAXA and NICT, and particularly remember Tobias Wehr, EarthCARE Mission Scientist, who passed away suddenly and unexpectedly in 2023, after more than a decade of support to the mission.



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
