# Peer review of "The EarthCARE Mission: Science Data Processing Chain Overview"

_EGUsphere, 2023_

## Author Response (AR1)

**Summary of relevant changes in manuscript version 3**

- Following a comment from reviewer 2, I moved the sections on product conventions and format up from the end of the paper into section 2, so that the reader is prepared for the product names in the processing chain description.
- The data flow and share of tasks between ESA and JAXA is described in more detail. Where descriptions were applicable to ESA only, this is now clearly indicated.
- A figure has been added to describe the overall processing flow together with the main parameters on each level (Figure 1).
- The description of the ATLID processing concept has been clarified, and an equation has been added (eq. 1).
- Sections 5.1.2 and 5.2.2 on the JAXA level 2 products were rewritten, to be in the same overall style as the corresponding ESA sections 5.1.1 and 5.2.1 (continuous text instead of lists).
- What is now Figure 3 has been replaced with a higher resolution version.
- A list of acronyms has been added, as requested by two of the reviewers.
- The list of references has been updated to reflect the current status of manuscripts for the AMT EarthCARE special issue: many papers have moved from preprints to final.
- Almost all detailed comments by the three reviewers were implemented.
- English and punctuation has been corrected in many places.

**Reply to Reviewer 1 comments**

Specific comments:

   Although it is understandable, this manuscript is filled with acronyms. It can be tiring to search where an acronym is introduced. Please add an appendix with the list of acronyms.

ME: Done

At an early place in the manuscript, e.g. in Sect. 2, please discuss the data distribution and timeliness of EarthCARE data: NRT, OFFL, direct broadcast? Which are the delivery means to users?

ME: This is addressed in the section 8.1 of the companion paper by Wehr et al. (https://amt.copernicus.org/articles/16/3581/2023). I added a sentence in section 1 (Introduction) to clarify this.

46: "…. which make use of meteorological data from ECMWF": does that also hold for the JAXA products?

ME: Yes, the same auxiliary data (ECMWF forecasts and a joint spatial grid) are also used by JAXA for the generation of their products. I added a note at the start of section 4 (Auxiliary data (level 1d) processors and products) to clarify this.

   2: It is unclear how the two processing chains of ESA and JAXA are aligned in time.

ME: I added a paragraph in what is now section 2.1 explaining the interaction between the two processing chains (second paragraph in section 2.1, starting "ATLID, MSI, and BBR level 1 products are generated at the ESA PDGS, then shared with the JAXA ground segment."

Figure 1:

The small font is poorly readable.

ME: This should be addressed by a high-resolution version of the figure in

the online version of the manuscript. Figure is now Figure 2.

Please mention the version, date etc. in the caption instead of in the top right corner. Is this diagram still up to date more than 5 years later?

ME: I updated the diagram to the latest version (version 8, 13 Sep 2023). Version and date are now listed in the caption as well.

Please mention the meaning of X-JSG and X-MET in the caption.

ME: Added to text explaining the diagram (not the caption).

Are all the acronyms explained somewhere in this paper?

ME: Yes, they are the ESA product names, listed and explained in Tables 1 and 3,

together with references to the detailed papers in this special issue.

Figure 2: It would be clearer to use the same instrument colour codes in Fig. 1 and Fig. 2. Is there a version/date of this JAXA flow diagram?

ME: Comment understood, however we prefer to keep the specific colours for the JAXA diagram. It is an up-to-date version (2023, no version number).

4.2: Please provide a figure with an example of the X-JSG grid.

ME: Figure has been added (Figure 4).

Technical points:

- Often text is put between brackets, but this is not always needed or allowed, and it hampers the readability. Please provide good running sentences instead.

ME: Text has been updated accordingly

- l. 2: global profiles? you probably mean: vertical profiles

ME: Confirmed, text updated

- l. 6: please mention the four instruments; this information belongs in the abstract

ME: Text updated

- l. 20: Explorer > Earth Explorer

ME: Text updated

- l. 83: Fig. (Figure) should be written in full if it is the first word in the sentence. The same holds for other words like Tab. and Sect.

ME: Text updated

- l. 95: "The differences in the Standard Product and the Research Products …" > "The differences between the Standard Products and the Research Products…" . Also at other places in Sect. 2.2 the English text and especially the singular/plural should be checked.

ME: Text updated

- L. 98 / l. 103: in > of

ME: Text reworded

- L. 100: indentation is missing

ME: now using table instead of two lists

- Sect. 3.1: what do you mean with: particles' signature, and molecules' signature? Please clarify.

ME: Text reworded

- L. 127: What about the effects of the instrument optics?

AND

- The entire paragraph from l. 125 to l. 130 about the signals and the optical chain is unclear. Please avoid using text between brackets, and reformulate the text with clear sentences. It could help to add an equation to explain the ATLID signal and its data processing.

ME: Paragraph completely rewritten, and equation added

- L. 139: ECGP: meaning?

ME: now explained when first used (first par. of section 2.1), and in acronym list

- L. 145: singular: detector?

ME: corrected

- L. 151: Is the A-NOM product already calibrated using the information from the three calibration modes/products mentioned below?

ME: yes, now explained in a general comment at the start of section 3

- L. 163: invalid raw data: you probably mean valid? the validity of invalid data is a contradiction.

ME: text reworded

- L. 168: gives

ME: corrected

- L. 170: please use a space between number and unit (km). This occurs at more places.

ME: corrected

- L. 185: detection raw data > the raw data

ME: corrected

- L. 186: slow > slowly

ME: corrected

- L. 199: please remove the brackets around "and the profile…" because this is important information.

ME: corrected

- L. 206: processing

ME: corrected

- L. 249ff: In the CPR … . Please check the use of articles in Sect. 3.2. Please check the syntax of the CPR text by a native English speaker. Often articles are missing and singular/plural is incorrect.

ME: corrected

- L. 249: Are all of these CPR products a function of height?

ME: no, some are per profile. Please refer to the Product description for details

- L. 258 ff: Is Pr a symbol or an acronym? Symbols (like Z) should be in italics, in equations and in text. Acronyms should be in upright font, in equations and in text.

ME: Pr is a symbol (reflected power). Changed symbols to italic font.

- L. 266: ….by surface estimation program written…: unclear

ME: reworded

- L. 269: delete: speeds

ME: done

- L. 270: what is IQ?

ME: Abbreviation spelled out and function of this detector explained

- L. 271-274: please reformulate

ME: done

- L. 274: subscript sat in upright font

ME: done

- L. 289: ; > :

ME: done

- L. 293: Have the calibration products M-DRK etc. mentioned below already been used in the processing of the Nominal L1B data product?

ME: yes, now explained in a general comment at the start of section 3

- L. 394: Why NB? Is this a footnote? A short sentence in the main text is preferable.

ME: "NB" removed

- L. 397: NEdT: please explain

ME: done

- L. 402: is LW = TW - SW?

ME: conceptually yes, but this is not true starting from measured TW and SW signals, which are modified by instrument effects (e.g., spectral response)

- L. 407: folding mirror's

ME: corrected

- L. 413: This product BM-RAD is not described below. Why is it called unfiltered whereas it should be a corrected product?

ME: Text revised to make this clearer. There's a bit of jargon: the L1 product is still convolved with the instrument response (or "filter") function, hence "filtered". Removing this effect in the L2b processing creates the "unfiltered" or "true" radiances, without the instrument effect.

- L. 430: Please explain directly below what you mean with marshalling.

ME: done

- L. 459: blackbody

ME: corrected

- L. 464: explain BB

ME: spelled out

- L. 468-469: product names B-LIN and B-SOL should be in bold, I guess.

ME: no, they are only in bold where the L1 products are introduced

- L. 472: this

ME: corrected

- L. 475: barycentres

ME: corrected

- L. 476 ff: too much text between brackets - check if this is needed.

ME: text between parentheses is redundant, removed

- L. 524: … two, two-dimensional (2D) grids > … two 2-dimensional (2D) grids

ME: corrected

- Next line: three dimensional > 3-dimensional

ME: corrected

- L. 528: The MSI grid: is this a 1 km x 1 km grid? is the fixed grid rectangular in lat/lon ?

ME: clarified in text - grid is 1 km x 1 km, and rectangular along/across track, not in lat/lon.

- L. 566: Tab. > Tables. Please do not use an abbreviation as the first word in a sentence

ME: updated in L. 556 and 559

- L. 576-578: unclear sentence.

ME: sentence reworded

- Sect. 5.1.2. and Sect. 5.2.2: please order these L2 products according to their instruments. Now they are mixed.

ME: Not changed. The order of products is the same as in tables 2 and 4. This is to separate standard and research products. Within each of these two groups, products are indeed ordered by instrument.

- L. 599: reformulate

ME: sections 5.1.2 and 5.2.2 been completely rewritten

- Table 1: This does not look like a logical ordering of products. Same comment for the other tables.

ME: Not changed. Products are grouped by instrument, then processor, following the production model left to right, and top to bottom, i.e., products produced first are listed first.

- Tables 1-4: Please put the table caption above the table. Please mention in the caption: single instrument L2 products or multi-instrument synergy products.

ME: done

- L. 629: CPR-ATLID > the combination of CPR and ATLID

ME: text updated

- L. 632: ACM_CLP > AC_CLP

ME: text updated

- L. 632: CPR-ATLID-MSI > the combination of the three instruments …

ME: text updated

- L. 636: what is 2D ice?

ME: text updated

- L. 647: the same

ME: text updated

- L. 665: EarthCARE production model

ME: text updated

- L. 684 ff: please give references for these software tools

ME: not done, to avoid including links which can become obsolete

- L. 727: delete: (4)

ME: text updated, now reads "(Sect. 4)"

- L. 754: plural: parameterisations, uncertainties

ME: text updated

**Reply to Reviewer 2 comments**

General

The title "Science Data Processing Chain Overview" implies that the manuscript will discuss the chains of the data processes, and how one data process affects another. However, the manuscript does not emphasize this perspective well. The manuscript mainly discusses the level 1 process for ATLID, CPR, MSI, and BBR in Section 3, followed by a brief description of the level 2 process. Therefore, it is not clear whether the manuscript's main point is the calibration processes used for the EarthCARE mission. If so, the title should change accordingly. Also, please check and revise the titles of subsections since some titles do not imply what is contained in the section.

ME: It is true that a significant part of the paper discusses level 1 processing and calibration. Apart from the companion overview paper by Wehr et al., this is the only place in this EarthCARE special issue where this is done, so it is necessary and required by the science user community to understand the full processing chain. For the level 2 processors on the other hand, there are individual papers in the same special issue. The aim of this paper is to put them into overall context, show what is available, and guide the reader to the detailed papers. That's why the level 2 section is comparably brief. The title reflects a convention at the space agencies where science data are the data from satellite instruments which are used to generate scientific (geophysical) data products. In this sense, Level 1 data are science data as well.

Section 2 indeed gives some descriptions of the processing chain performed by ESA and JAXA. However, the first two figures (Figs. 1 and 2) would not be very informative to most readers unless sufficient description is provided in there. Many abbreviations used in these two figures are not explained in Section 2. Later sections, such as section 7.2, actually give information about the processor names and which parameters are contained in the product. Therefore, it would be better to rearrange sections to give more information about the names of instruments, algorithms, and processors followed by the figures about the chains of algorithms. Moreover, the two agencies use quite different ways of product naming conventions (ESA uses "A" for ATLID and JAXA uses "ATL"). Providing a list of abbreviations or tables would be very helpful to readers.

This manuscript is a great output from multiple agencies' achievements, but I feel that the manuscript can be more coherent if sections are reordered and merged properly. For example, it would be better if a brief description is given about the names of instruments, processors, and products. Then provide how the two agencies collaborate, which agency performs level 1 processing and level 2 processing, and where the data product will be provided. Then it would be better to discuss how the algorithms/processors are connected in a chain process. To show the chain, it would be nice which parameter is passed from level 1a to level 1b, or level 1 to 2, etc. For example, attenuated backscatter produced in level 1 is used in level 2 ATLID, and CPR radar reflectivity produced in level 1b is used in level 2 CPR processing. Figures 1 and 2 do not contain such information. After that, a specific description can be provided for level 1a/b/c/d and 2a/b algorithms. Sections 6 and 7 also contain important information. The simulator discussed in Section 6.1 might be provided in the overview of algorithm chains and development.

ME: To address your comments, I added figure 1 outlining the overall processing flow and providing the main parameters for each block. I updated what is now section 2.1 to clarify the respective responsibilities of ESA and JAXA. I moved the former section 7 (Product format and conventions) into section 2 (as section 2.2), in order to introduce processor naming conventions before they are used, and added a list of acronyms at the end of the paper. I kept section 6.1 on the end-to-end simulator where it is, together with the other development related sections, as I feel it would break the flow moving it up to section 2.

Specific

L111: It seems that the 3 optical paths refer to the path within/inside the instruments. If so, please mention it.

ME: reworded for clarity

What does the "processor" mean? How different is it from the algorithm, computer resource, or the science team? Sometimes, the sensor name was used as the algorithm development team in the manuscript, and it is confusing.

ME: now explained in section 2.3

Section 3: The information about the calibration processing is interesting. However, if the manuscript's main focus is the processing chain, the specific description provided in Section 3 is not directly connected to the main point. Please refine this section to be coherent with the main topic.

ME: see my comment above

L565: Is this cloud climate product one of the Research Products?

ME: ESA does not have the concept of Research Products. The cloud climate product CLIMP will not be produced in the ESA ground segment, but directly and ad-hoc by its developers.

Section 6.4 contains important messages for the algorithm developers.

ME: noted

L702: It is not clear what the limitations refer to.

ME: examples added

L 708: Please clarify "agile developments".

ME: clarification added in the text. This refers to agile software development practices. see, e.g., https://en.wikipedia.org/wiki/Agile_software_development

**Reply to Reviewer 3 comments**

The manuscript provides a through overview of the data processing chains for data downlinked by the multiple instruments onboard the upcoming EarthCARE satellite and information on the calibration processes for each instrument. Given the topic, it is entirely suitable for consideration by EGUSphere. All together the manuscript is organized logically and written clearly. There is sufficient detail provided to understand the data processing chains that are described. The calibration algorithms are described well too, though it would be helpful to let the reader know if this document is intended to be the primary calibration documentation or if there are other publications that go into further detail. If the latter is true, then it is recommended that the relevant citations be added. I only have a small number of minor questions beyond the excellent questions provided by Reviewers #1 and #2. Due to the strength of the manuscript, I recommend publication after minor revisions.

ME: comment added at start of section 3 to indicate that this (together with the companion paper by Wehr et al.) is indeed the primary source of information on calibration of EarthCARE instrument

Figure 1. The font size is small, but it is necessary. Recommend submitting a higher resolution version for the published manuscript so that the words are legible.

ME: will do

Section 3. A lot of statements are in parentheses which makes the many sentences difficult to read. This is a stylistic choice, so it is not wrong, but it may be worth reconsidering if some sentences would be easier to read and more direct with by re-wording without the parentheses.

ME: revised, removing a lot of the sentences in parentheses

Line 153. It is not clear what is meant by 'segregated-type associated errors'. I think this phrase is trying to simultaneously describe multiple things. Please clarify or consider re-wording if appropriate.

ME: reworded, removing segregated-type

Line 165. What is meant by measurement validity bit? Is this a bit that is reported in a QA flag? It may be clearer by rewording this to "as indicated by the measurement validity bit in xxx flag" or similar.

ME: removed "measurement validity bit" as it does not add anything

Lines 165-166. Please explain what the control loop is or maybe provide a reference to documentation that explains where this information can be found. Also, it is not clear how 'co-alignment emitter/receiver' is related to the control loop status. Adding more information here would help the reader.

ME: clarified that this is the laser beam steering control loop. For more detail see section 5.1 in the companion paper by Wehr et al., specifically pages 3587/3588.

Line 210. "The baseline method for inferring the Rayleigh cross-talk value…". Is there a reference that describes the details of the baseline method or is it being introduced here? If this is an established method, then it is recommended to add a citation so the reader can understand the details and assumptions of the method. If this is the first publication of the baseline method, then the amount of information and justification provide in Section 3.1.4(a) seems to little for a rigorous understanding.

ME: this is an outline of a new method which has just been developed and not yet been published. Which is why we can neither provide more detail here nor provide a reference.

Line 213. Is there a notional SNR value that is 'sufficient' that should be added or is this premature to add to this publication?

ME: this is still premature as this is a new algorithm. It will be tuned once ATLID is in orbit.

Line 222. The parenthetical item '(estimation window)' makes the sentence difficult to read. It would be better to revise as something like, 'called the estimation window'.

ME: updated

Line 231-237. The method would be more clear if the channel duplexing operations (b) and (c) included equations that shows the form of the correction matrix and for the computation of the cross talk corrected signals. The basic idea is communicated, but if this manuscript is intended to be a reference then these details seem important. Alternately, if there is another publication or algorithm documentation with these details, a citation is an appropriate solution.

ME: Equation (1) added which shows the principle. There will be more information in a planned publication.

Line 239. Where do the lidar absolute constants come from? Also, it would be clearer to explain how these calibration constants are applied. Are they multiplied, divided,..etc? Adding an equation here might clear this up.

ME: See equation (1) for the application. Lidar constants are derived on the fly from pure Rayleigh scattering targets (clean stratosphere). This will also be covered in the planned publication. Apologies for not being able to provide more details here as the algorithm is still very new.

Line 397. NEdt is undefined.

ME: now spelled out

Line 695. The parenthetical item 'fundamental' makes this sentence awkward to read. Consider revising.

ME: reworded

Line 698. 'Expertise and developments should be organized vertically…'. Why is this so? Please add a reasoning for this assertion.

ME: reasoning added